# Galectin-3 in Kidney Diseases: From an Old Protein to a New Therapeutic Target

**DOI:** 10.3390/ijms23063124

**Published:** 2022-03-14

**Authors:** Louis Boutin, François Dépret, Etienne Gayat, Matthieu Legrand, Christos E. Chadjichristos

**Affiliations:** 1FHU PROMICE AP-HP, Saint Louis and DMU Parabol, Critical Care Medicine and Burn Unit, AP-HP, Department of Anesthesiology, Université Paris Cité, 75010 Paris, France; louis.boutin@inserm.fr (L.B.); francois.depret@aphp.fr (F.D.); etienne.gayat@aphp.fr (E.G.); 2INSERM, UMR 942, MASCOT, Cardiovascular Marker in Stress Condition, Université Paris Cité, 75010 Paris, France; matthieu.legrand@ucsf.edu; 3Department of Anesthesiology and Peri-Operative Medicine, Division of Critical Care Medicine, University of California—UCSF Medical Center, 500 Parnassus Ave, San Francisco, CA 94143, USA; 4INI-CRCT Network, 54500 Nancy, France; 5INSERM UMR-S1155, Tenon Hospital, Sorbonne Université, 75020 Paris, France

**Keywords:** galectin-3, kidney disease, lectins

## Abstract

Galectin-3 (Gal-3) is a 30KDa lectin implicated in multiple pathophysiology pathways including renal damage and fibrosis. Gal-3 binds β-galactoside through its carbohydrate-recognition domain. From intra-cellular to extra-cellular localization, Gal-3 has multiple roles including transduction signal pathway, cell-to-cell adhesion, cell to extracellular matrix adhesion, and immunological chemoattractant protein. Moreover, Gal-3 has also been linked to kidney disease in both preclinical models and clinical studies. Gal-3 inhibition appears to improve renal disease in several pathological conditions, thus justifying the development of multiple drug inhibitors. This review aims to summarize the latest literature regarding Gal-3 in renal pathophysiology, from its role as a biomarker to its potential as a therapeutic agent.

## 1. Introduction

Galectin-3 (Gal-3) is a lectin discovered in the early 80s in tumoral cells. Since, the role of this protein has been studied in several organs and associated injuries [1,2,3]. Whether it is cancer, cardiovascular, or renal diseases, its pathophysiology remains complex. However, Gal-3 pharmacological inhibition could have potential therapeutic implications. The involvement of Gal-3 in renal diseases has been recently explored from basic science to clinical trials, and its increased expression can be associated with specific renal damage and prognosis.

The objective of this review is to discuss Gal-3 pathophysiology in kidney disease as a biomarker and a potential therapeutic target.

## 2. Gal-3: A Carbohydrate Binding Protein from the Lectin Family

### 2.1. The Lectin Family

The term “lectin” is derived from the Latin word “legere” which means “to collect” and defines a group of binding proteins that interact with multiple partners via a specific carbohydrate recognition domain (CRD) [4]. Because of these interactions, lectins are implicated in a wide variety of pathways (e.g., from transduction pathway to cell-to-cell interaction) [5,6].

Galectins are a 14-member family of proteins within the lectins that bind β-galactose via their specific CRD. Galectins are divided into three groups, depending on their CRD and protein structures: galectins 1, 2, 5, 7, 10, 11, 13, 14, 15, and 16 are composed of one unique CRD associated in monomer or in dimer; galectins 4, 6, 8, 9, and 12 are composed of two different united CRDs; and Gal-3 is a chimeric protein including one CRD and one regulatory N-terminal domain connected with a collagen-like sequence [7]. Galectins’ architecture is described in Figure 1a. Galectins are involved in various biological processes such as early development, cell migration, immunological signaling leading to profibrotic effects [8,9,10], and cell-to-cell communication [11].

### 2.2. Galectin-3

#### 2.2.1. Structure

Gal-3 is the only galectin discovered specifically in mammals and has been renamed many times (Mac-2 antigen, IgE-binding protein, carbohydrate-binding protein, and L-29) [12,13,14]. It is a soluble protein composed of 3 distinct domains: a CRD, a collagen-like sequence, and a specific N-link terminal domain [15,16,17,18]. The structure of galectin-3 is described in Figure 1b.

The CRD is a polypeptide fold domain binding carbohydrates. Gal-3’s CRD interacts with various carbohydrate-containing proteins, activating different signaling pathways [19].A collagen-like sequence links CRD to N-link domain and is composed of nine collagen-like sequences (proline/glycine rich domain) cleavable by matrix metalloprotease [20].N-link domain: it is essential to Gal-3 biological activity. This domain has two sites where serins are phosphorylated [21]. Biochemical modifications of this domain lead to Gal-3 internalization in kernel cells [22].

#### 2.2.2. Expression and Role

Multiple organs (i.e., lung, spleen, stomach, colon, adrenal glands, uterus, ovarian, and kidney) and diverse cells (i.e., inflammatory, endothelial, muscle or tumor cells, and fibroblasts) express Gal-3, leading to different roles in various pathophysiological conditions [23,24,25].

Expression of Gal-3 can be intracellular, on the cell membrane, or in a soluble form, outside the cell. A brief description of Gal-3’s various roles is described in Figure 2.

Inside the cell, Gal-3 is formed in the cytoplasm and promotes cell survival via apoptosis regulation [26]. It can also be internalized into the nucleus to promote cell proliferation [27].Gal-3 is found in the plasma membrane of cells where it can modulate the interaction between epithelial cells and extracellular matrix or with other cells [28]. Its ability to bind with integrins or endothelial adhesive proteins allows adhesion with other cells or the promotion of activation of bond cells [29,30,31,32]. Gal-3 can also bind with glycoproteins to promote extra-cellular matrix binding [28,33] and create a bridge between Gal-3 and cells [34]. All these mechanisms promote a transduction cascade from membrane to intra-cellular pathways [22,35,36,37,38].Gal-3 is also secreted in plasma, or in organs in a soluble form, during specific damage, and acts as a damage-associated molecular pattern (DAMP) to promote an immunological response [39,40]. Many studies reported that Gal-3 plays a profibrotic role in kidneys and lungs via immune modulation of macrophage infiltration [41,42,43].

#### 2.2.3. Galectin-3 Inhibitors

Genetic and pharmacological inhibition of Gal-3 has been investigated to evaluate its pathophysiological impact after injury. Since this lectin binds to multiple cellular sites, extracellular fixation and permeability abilities are crucial for Gal-3 inhibitors, which are classified according to their carbohydrates’ binding characteristics [44]. Gal-3 inhibitors currently used are listed in Table 1. However, some trials assessing Gal-3 inhibitors remain incomplete in their approach and relevance. In this matter, the use of Gal-3 inhibitors in clinical assays has limited efficiency. Additional studies are needed to investigate this point.

#### 2.2.4. Galectin-3 in the Kidney

During development, Gal-3 is expressed mainly in collecting ducts, within or on the apical membrane of α-intercalated cells. This suggests a role of Gal-3 in tubular development [54], possibly via cell-to-cell adhesion or interaction with the extracellular matrix to favor tubulogenesis [55]. In the adult kidney, Gal-3 expression is detected in principal and intercalated cells, in proximal tubules, and the ascending thick limb [56].

Pharmacokinetics of Gal-3 are not fully understood, but many exploratory works suggest a renal elimination as it can freely pass through the glomerular membrane [27]. Gal-3 is a low molecular weight protein, between 29 to 31 KDa, and there is no evidence showing an association with albumin in extra-cellular compartments [57]. Meijers et al. first evaluated in a rat model experiment that Gal-3 was freely and completely excreted in urine with a clearance of 0.92 mL/min and a volume of distribution of 90 mL. In healthy humans, the calculated Gal-3 excretion rate was 3.9 mL/min (2.3 to 6.4) and fractional excretion of Gal-3 was 3.0% (1.9 to 5.5). Finally, for patients in hemodialysis, Gal-3 can be filtered by the dialyzer, but with a lower rate than creatinine [58].

In pathological conditions, Gal-3 expression varies and has been studied in preclinical and clinical studies. The following paragraphs will detail these studies.

## 3. Gal-3 in Preclinical Models of Kidney Disease

The role of Galectin-3 is summarized in Figure 3.

### 3.1. Ischemia/Reperfusion

The role of Gal-3 in renal ischemia/reperfusion (rIR) has been reported in various studies.

Nishiyama et al. showed that rIR in rats induced a rapid Gal-3 mRNA overexpression and had a negative correlation with serum creatinine dosage at 48 h (R = −0.94). Gal-3 overexpression was extended in distal tubules 48 h after the injury [59]. Furthermore, genetic inhibition of Gal-3 in this model was associated with less macrophage infiltration and activation, less tissue damage (ROS production, tubular necrosis), and improved renal function [60].

We recently reported an increase of plasmatic, mRNA, and protein expression of Gal-3 24 h after rIR due to tubular destruction and inflammatory cells infiltration. Increased plasma levels of pro-inflammatory cytokines, such as IL1b, IL6, TNFα, and IL10, were associated with secretion of Gal-3 by immune cells and correlated to plasma soluble Gal-3 overexpression. Interestingly, increased levels of pro-inflammatory cytokines were blunted in Gal-3 KO mice [61]. Moreover, a population of renal interstitial cells Gal-3 (+), CD44 (+), and vimentin (+) (markers of regeneration processes) was identified in the outer medulla around necrotic tubules and in adjacent capillaries after rIR, suggesting that Gal-3 mediates endothelial activation and kidney tubular regeneration in these experimental conditions [62].

All these studies revealed a specific role of Gal-3 after rIR, promoting endothelial activation, cytokine secretion, and renal inflammation, leading to tissue remodeling and further fibrosis. In this model, inhibition of Gal-3 protected from renal disease-associated consequences (inflammation, fibrosis, cytokine secretion) [63].

### 3.2. Toxic Injury

In toxic preclinical models of experimental nephropathy, tubular damage appears less pronounced than in rIR models. In a toxic model of nephropathy induced by folic acid (FA), Nishiyama et al. observed a rapid increase of renal Gal-3 mRNA expression [59]. Furthermore, a spread expression of Gal-3 from proximal ducts to a diverse subset of tubules, including dilated collecting ducts, was observed. Fourteen days after injury, Gal-3 was also detected in macrophages and fibrosis progression was prevented after Gal-3 inhibition with less renal apoptosis and inflammation [64].

In a cisplatin-induced AKI model, Li et al. observed an increase of Gal-3 renal expression at day 3, associated with an overexpression of PKC-α, cell apoptosis, and collagen type I synthesis. Interestingly, Gal-3 inhibition limited the AKI to CKD transition [65]. In contrast, Volarevic et al. demonstrated, in a similar model of AKI, that Gal-3 expression in immune cells was associated with regulation of immunosuppression via renal dendritic cells, TLR-2 activation, and IL-10 secretion. These findings suggested a possible deleterious role of Gal-3 inhibition for immune regulation [66].

Thus, Gal-3 may play contradictory roles, depending on the cell that expresses it after kidney injury (i.e., pro-fibrotic when it is synthesized by tubular cells, and anti-fibrotic when produced by macrophages).

### 3.3. Glomerular Injury

Altered expression of Gal-3 within injured glomeruli has been reported in some studies. In a streptozotocin model of diabetic nephropathy in rats, Gal-3 was overexpressed from 2 to 12 weeks in diabetic rats, and mainly in mesangial cells. Overexpression of Gal-3 modulated the glomerular remodeling of associated advance-glycation-end-product (AGE) receptor (RAGE) [67]. Using the same experimental model, these authors demonstrated that Gal-3 deficiency resulted in accelerated diabetic glomerulopathy accompanied by glomerular AGE accumulation due to RAGE downregulation [68]. In addition, injection of N-carboxylmethyllysine in mice induced AGE accumulation associated with the glomerular injury. Furthermore, Gal-3 inhibition leads to an accelerated glomerular disease via higher circulating AGE levels and altered RAGE functions [69]. Zhang et al. proposed a mechanism of AGE-mediated damage via a long non-coding sequence Rpph1 interacting with Gal-3, promoting MERF/ERK transduction pathway resulting in MCP-1 overexpression and mesangial cell proliferation [70].

In another preclinical model using transgenic Ren-2 rats which developed severe hypertension-associated glomerulosclerosis, the progression of glomerulopathy and related proteinuria was associated with increased plasma Gal-3 levels. In this model, inhibition of Gal-3 improved both renal damage and function [71].

Finally, injections of anti-Thy1.1 antibodies in rats showed that the progression of glomerulonephritis was associated with increased expression of Gal-3 in distal tubules and in infiltrated glomerular macrophages and may play a role in mesangial hypercellularity [72].

These data demonstrate the complex role of Gal-3 in glomerular damage. Inhibition of Gal-3 seems to downregulate its reduction leading to acceleration of glomerulopathy, whereas it may play a protective role when its expression is extending in tubule and in mesangial cells during glomerulopathy, leading to a specific Gal-3-associated immune cells infiltration reduction.

### 3.4. Immune-Associated Renal Damage

#### 3.4.1. Sepsis-Associated Renal Disease

In a peritonitis model in rats (cecal ligature puncture), Gal-3 was upregulated in septic-associated renal damage. Furthermore, pharmacological inhibition of Gal-3 using modified citrus pectin (MCP) decreased IL-6 plasma and renal inflammation and improved renal function and survival [73].

#### 3.4.2. Transplantation Model

Gal-3 has been shown for several years to modulate inflammation and immune cell infiltration in pathophysiological conditions. Grafted models are associated with immune cell-related graft dysfunction. Dang et al. grafted BM12 kidney from WT mice, in WT mice or in Gal-3 null mice. In WT mice, this graft resulted in Gal-3 tissular and plasmatic upregulation. Gal-3 null mice had less tubular injury, moderate fibrosis, and a reduced amount of immune cells infiltration [42]. These studies confirmed the role of Gal-3 in the recruitment of immune cells during the pathological context, and its inhibition improved renal outcome.

### 3.5. Polycystic Model

In a model of congenital polycystic kidney (CPK) in mice, Gal-3 expression was observed in cilia of dilated collecting ducts. Interestingly, injection of Gal-3 decreased the cyst number in mice whereas Gal-3 null mice showed a higher kidney weight/bone length ratio and a modified cilia structure.

This study underlines the major role of tubular Gal-3 expression in the development of the tubule and its structural consequences [74].

### 3.6. Renal Fibrosis

Gal-3 has been reported to promote fibrosis in several organs [1] and plays a major role in the renal transition from acute to chronic disease, promoting inflammatory factors release, inflammatory cell activation, and tissue injury [75]. However, the role of Gal-3 in this purpose is still a matter of debate.

Gal-3 is associated with the proliferation of extra-cellular matrix-producing cells (fibroblast and myofibroblast), which may be in turn associated with the migration and adhesion of such cells [76].

It has also been reported that the degree of renal damage and fibrosis was more extensive in Gal-3 null mice with increased total collagen, whereas a corresponding decrease of myofibroblast and extra-cellular matrix synthesis via a downregulation of endo180 receptors implicated in collagen degradation was observed [77].

Using a model of unilateral ureteral obstruction (UUO), Gasparitsch et al. demonstrated that Gal-3 expression was associated with an increased expression of collagen I. The authors observed that Gal-3 expression was less important in RAGE−/− or RAGE−/− ICAM−/− mice. These data suggest that the association of Gal-3 and RAGE / ICAM is involved in the transition from acute damage to fibrosis [78]. In the same experimental model, Henderson et al. focused on macrophage infiltration and observed that macrophages that expressed or secreted Gal-3, promoted renal fibroblast activation to a profibrotic phenotype. Furthermore, specific depletion of macrophages using CD11b-DTR mice reduced fibrosis severity in this model [63].

Finally, using a high fat diet in rats, a model characterized by pronounced glomerular and tubular damage, pharmacological inhibition of Gal-3 improved immune response and renal fibrosis. In addition, in another model of fibrosis induced by aorta occlusion, pharmacological inhibition of Gal-3 also alleviated fibrosis [79].

These studies demonstrate the pro-fibrotic role of Gal-3 expression in pathophysiological response to injury. Expression of Gal-3 in immune cells seems to be associated with renal fibrosis. However, expression of Gal-3 in tissue leads to better tissue regeneration and less fibrosis following renal injury through collagen regulation via specific receptors.

### 3.7. Preclinical Model of Cardio-Renal Syndrome

We have recently identified the potential role of Gal-3 in type 3 cardio-renal syndrome. In this study, mice were operated using left renal ischemia reperfusion after right nephrectomy to induce a transient renal dysfunction, rapidly normalized at 48 h, inducing no water overload or no uremic syndrome. This leads to a decrease of cardiac fraction shortening and an increase of cardiac fibrosis after 28 days. Similar results were observed after unilateral ureteral occlusion. This study highlighted the role of acute renal damage leading to a beginning of cardiac dysfunction. In this experimental model, Gal-3 was increased after renal ischemia reperfusion in plasma, renal tissue, and later in cardiac tissue. The inhibition of Gal-3 prevented cardiac dysfunction. Furthermore, we identified in a model of bone marrow graft mice that immune cells expressing Gal-3 in the heart promoted cardiac fibrosis and its inhibition in bone marrow grafted cells improved cardiac phenotype. This study suggests a potential role of Gal-3 in crosstalk between the kidney and the heart during type 3 cardio-renal syndrome [61]. The role of Gal-3 in the initiation of cardiorenal syndrome remains unknown, some preclinical studies are trying to identify the renal role of Gal-3 in this context [80].

## 4. Gal-3 as a Biomarker

### 4.1. Kidney Function

Initially, Gal-3 was studied as a biomarker of cardiac injury [81], but since, several studies have evaluated its role as a biomarker of acute kidney injury.

Drechsler et al. measured Gal-3 baseline level from the German Diabetes mellitus Dialysis (4D) study (1168 dialysis patients with type 2 diabetes mellitus) and the Ludwigshafen Risk and Cardiovascular Health (LURIC) study (2579 patients with coronary angiograms). Gal-3 level gradually increased with the severity of renal function: from 12.8 ± 4.0 ng/mL (eGFR ≥ 90 mL/min per 1.73 m^2^) to 54.1 ± 19.6 ng/mL (dialysis patients of the 4D study) [82].

Gal-3 plasmatic value after cardiac surgery was studied by Von Ballmos et al. for AKI prediction in 1498 patients and the highest tercile of Gal-3 was associated with severe AKI (OR of 2.95; *p* < 0.001) [83].

In a long term follow up study of 1320 patients with type 2 diabetes and an eGFR ≥ 30 mL.min^−1^ 1.73 m^−2^, Tan et al. demonstrated that Gal-3 was independently associated with doubling of serum creatinine (HR 1.19 CI_95%_[1.14, 1.24], *p* < 0.001) even after adjusting for chronic renal risk factors, baseline eGFR, and albuminuria status [84].

In another translational study including patients admitted in ICU with severe sepsis, it was found that median serum and urine Gal-3 levels at admission were higher for patients with AKI (AKI vs. non-AKI serum: 18.37 vs. 8.08 ng/mL, *p* < 0.001; AKI vs. non-AKI urine: 13.27 vs. 6.27 ng/mL, *p* < 0.001), with a good prediction performance for AKI (AUC of 0.88 for serum Gal-3 and 0.87 for urine Gal-3) [73]. In a population of heart failure, Tariq Ahmed et al. explored a cohort of 132 patients, and urinary Gal-3 was associated with an increased risk of death after adjustment on a renal injury biomarker (urinary NGAL) [84].

More recently, we have reported that plasma Gal-3 level at ICU admission, with heterogenous diagnosis, was found to be associated with AKI with an OR of 1.12 CI_95%_[1.04, 1.2] after adjustment on severity biomarkers and non-renal recovery confounding factor (i.e., gender, age, CKD, vasopressor treatment, SAPSII, Charlson score, Screat at admission, and lactate value at admission). Plasma Gal-3 increase was correlated with the severity of AKI, 16.6 (12.7–34.2) ng/mL for no AKI, and from 23.6 (18.2–34.2) ng/mL for KDIGO 1 to 38 (24.5–57.1) ng/mL for KDIGO 3 [85].

These studies suggest a potential use of plasma and urinary Gal-3 as a biomarker of the severity of AKI in a heterogenous population regardless of renal dysfunction origin.

### 4.2. Proteinuria

Kikuchi et al. reported a correlation between glomerular infiltrated Gal-3 positive monocytes and proteinuria in renal biopsies of 37 patients with glomerulonephritis (GN) (r = 0.616, *p* < 0.001) [86]. Moreover, in lupus GN, Kang et al. found a pronounced glomerular expression of Gal-3 in 81.8% (72/88) of patients associated with renal inflammatory cells. This overexpression of Gal-3 was correlated with histologic activity indexes, anti-dsDNA titers, and complement 3 and 4 levels [87]. In children, Ostalska-Nowicka et al. explored various types of GNs (minimal change disease (MCD), mesangial proliferation (DMP) and focal segmental glomerulosclerosis (FSGS)) and identified cortical and medullary Gal-3 positive cells highly expressed for no responding to steroid therapy (*p* < 0.001) [88].

Seventy-five patients with Mediterranean fever (FMF) with GN had higher serum Gal-3 levels compared to the control group and more importantly, for patients with proteinuria with a correlation ratio for proteinuria/creatinine of 0.785, *p* < 0.001. Prediction performance of serum Gal-3 for proteinuria had an AUC of 0.88 [89].

In a cross-sectional study including 90 patients, Gal-3 plasma level was significantly higher in macroalbuminuria (*p* ≤ 0.05) and for patients with poor kidney function (Stage IV–V CKD), with a prediction performance of 0.776 (CI_95%_[0.677, 0.875]; *p* ≤ 0.0001) [90].

Based on these studies, increased renal and plasmatic Gal-3 expression is associated with immune cells infiltration expressing Gal-3 in patients with glomerular injury.

### 4.3. CKD and Renal Prognosis

Alam et al. identified high plasma levels of Gal-3 in patients with severe comorbidities (heart failure, CKD) in 2 longitudinal cohorts, including patients with CKD: the Clinical Phenotyping and Resource Biobank (C-PROBE) study and the Seattle Kidney Study (SKS), which were associated (HR = 1.35, CI_95%_[1.01–1.80]) with chronic renal disease [91]. In a prospectively analyzed study from Atherosclerosis Risk in Communities (ARIC), Rebholtz et al. measured Gal-3 plasma levels in 9148 patients with no chronic kidney disease and no chronic heart failure. The authors showed that Gal-3 was higher for low estimated glomerular filtration rate, low urine albumin-to-creatinine ratio, and was associated with CKD with an OR of (2.22 CI_95%_[1.89, 2.60]) [92].

In another cohort including patients with chronic kidney disease, Gal-3 plasma levels were associated with elevated serum creatinine, urine protein/creatinine ratio, and were independently associated with CKD progression [93].

Our group explored renal prognostics using Major Adverse Kidney Event criteria in a cohort of 2076 patients admitted in ICU. We observed that Gal-3 dosage at admission was associated with MAKE with an OR of 1.37 CI_95%_[1.27, 1.49]. Predictive performance of Gal-3 for MAKE had an AUC ROC for predictive performance of MAKE of 0.76 CI_95%_(0.74–0.78) [85].

Ou et al. identified, in 249 patients who underwent kidney biopsy, that patients with CKD compared to those without had a higher level of plasma Gal-3. Higher Gal-3 levels were also associated with interstitial fibrosis, tubular atrophy, and vascular intimal fibrosis and RNA-sequencing analysis showed the upregulation of Gal-3 in fibrotic kidney biopsy samples [94].

These studies demonstrate the association and potential role of Gal-3 for poor renal prognosis and evolution of acute injury to chronic renal damage.

### 4.4. Transplantation

There is a major need for a good biomarker in kidney transplantation as it can allow therapeutic strategy changes. Sotomayor et al. analyzed 561 patients and baseline median Gal-3 of 21.1 (IQR [Q1:17.0, Q2:27.2] ng/mL. In this study, Gal-3 was associated with increased risk of graft failure (hazard ratios (HR) per 1 SD change, 2.12; CI_95%_[1.63, 2.75]; *p* < 0.001), more importantly for patients with hypertension (HR, 2.29; CI_95%_[1.80, 2.92]; *p* < 0.001) or smoking history (HR, 2.56; CI_95%_[1.95, 3.37]; *p* < 0.001) [95]

### 4.5. Mortality and Poor Cardiovascular Outcome

Finally, patients with renal diseases often have higher mortality rates. In 4D and LURIC, Gal-3 plasma high levels were significantly associated with all-cause mortality, or cardiovascular mortality, in a population with renal impairment compared to patients with no renal disease [82]. Hogas et al. confirmed in patients with hemodialysis and observed that a level of Gal-3 > 23.73 ng/mL was an independent predictor of mortality (HR: 2.60; CI_95%_[1.09, 6.18]) [96].

Later studies, in a big meta-analysis enrolling 5226 patients, Zhang et al. confirmed this association between Gal-3 and an increased risk of all-cause mortality and cardiovascular (CV) event in CKD patients (HR:1.379, [1.090, 1.744]) and associated with the risk of CV events in CKD patients (HR = 1.054, CI_95%_[1.007, 1.102])[97].

Thus, Gal-3 has been demonstrated to be a good biomarker for poor cardiovascular prognosis, and more precisely in patients with AKI [98]. These results highlight the role of Gal-3 in cardio-renal syndrome as it is associated with poor renal and cardiovascular prognosis. Some preclinical studies have started to identify Gal-3 as a key player in the type 3 cardio-renal syndrome [61]. The mechanism in this context remains complex but associates both immune and tissue expression of Gal-3.

Finally, we recently identified, in a population admitted in ICU, that Gal-3 at admission was higher for non-survivor patients, was further associated with all-cause mortality at 30 days (OR CI 1.25, CI_95%_[1.17, 1.34]), and had a prediction performance of 0.69 (CI_95%_(0.67–0.72) [85].

Thus, Gal-3 appears to be associated with cardiovascular outcomes and mortality after kidney diseases and injury.

All associations of galectin-3 and outcomes in clinical studies are detailed in Figure 4.

## 5. Gal-3 as a Therapeutic Target and Perspective

Only few clinical studies evaluated the impact of Gal-3 inhibitors and their therapeutic implications. A phase IIa blinded, multi-center, randomized clinical trial that enrolled 121 patients with stage 3b or 4 CKD has been initiated using the Gal-3 inhibitor GCS100 (registered in Clinical Trial as NCT01843790, Table 1). This study aimed to evaluate the change in eGFR from baseline relative to placebo after administration of GCS-100 for 8 weeks [99]. In a press-release, the authors reported that 121 patients with chronic kidney disease were treated with a pharmacological Gal-3 inhibitor and significantly improved their glomerular filtration rate, uric acid, and blood urea nitrogen (BUN) levels, compared to placebo, between baseline and end of treatment. The authors did not report any severe adverse effects using 1.5 mg/m^2^ [100].

Lau et al. evaluated the use of MCP in hypertensive cardiac complications in a randomized controlled trial and found that inhibition of Gal-3 did not influence fibrosis cardiac biomarkers expression, but was slightly associated to a diminution of plasmatic creatinine and an increase of eGFR in MCP-treated patients. This suggests a potential interest of using Gal-3 inhibitors in patients with renal injury but data remain insufficient to draw definitive conclusions [101].

Gal-3 has also been evaluated as a treatment of fibrosis in pulmonary disease. Hirani et al. treated 36 healthy patients and 24 patients with pulmonary fibrosis with inhaled Gal-3 inhibitor and reported a good tolerance in healthy patients and a diminution of plasmatic markers associated with pulmonary fibrosis [102]. Some studies used Gal-3 inhibitors in the treatment of cancer, using drug resistance and survival endpoint [103], but they remain insufficient for therapeutic recommendations. Nevertheless, clinical data are missing to identify the impact of inhibition in clinical care and particularly for improving renal outcome. Preclinical data are promising, but the pathophysiology of renal protection remains unclear. Additional studies are needed to propose this treatment in a clinical point of view.

A deeper investigation of Gal-3 or related pathways has multiple perspectives. First, the use of Gal-3 as a biomarker is a major breakthrough for the clinical management of patients with renal disease and can help to monitor and guide specific therapeutic management. Second, even though its specific therapeutic use has not been established yet, Gal-3 can help stratify patients with renal damage. More preclinical studies are needed to confirm Gal-3 inhibition as a potential therapeutic target to limit renal injury.

## 6. Conclusions

Gal-3 has been explored in renal disease, from preclinical models to clinical studies. Basic science partially identified the pathophysiology of renal Gal-3 during specific renal damage and proposed its inhibition as a potential therapeutic target. Clinical work evaluated Gal-3 as a biomarker of renal disease to monitor and guide therapeutic strategies. To confirm Gal-3 inhibition’s potential benefits, some mechanistic studies are still needed.

## Figures and Tables

**Figure 1 ijms-23-03124-f001:**
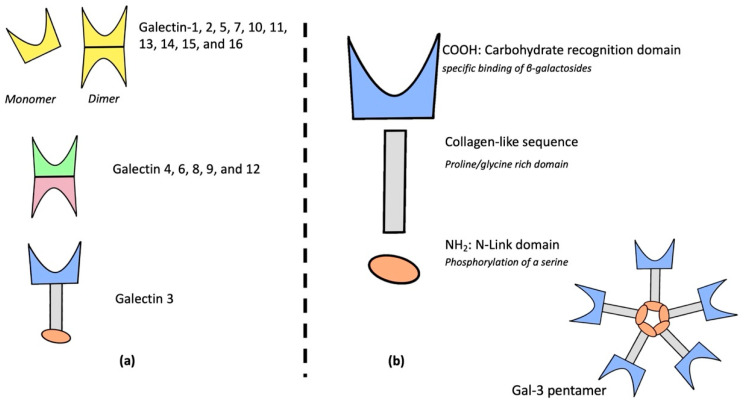
Galectins’ family structure (**a**), galectin-3 chimeric and specific structure (**b**).

**Figure 2 ijms-23-03124-f002:**
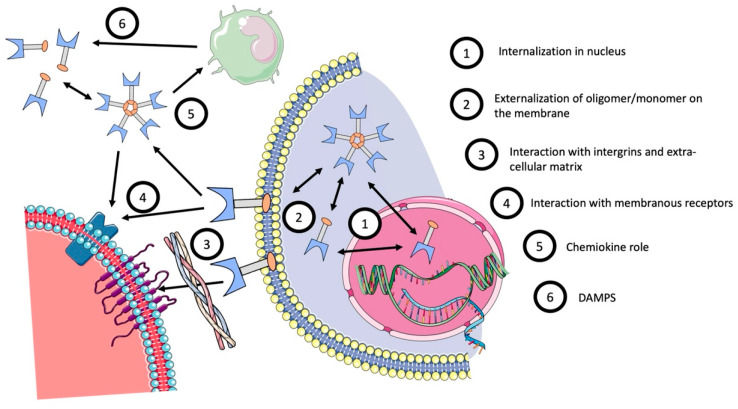
Galectin-3’s role and localization.

**Figure 3 ijms-23-03124-f003:**
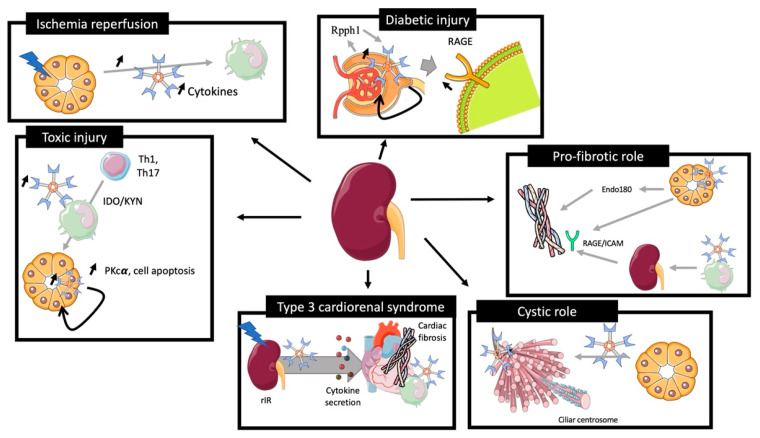
Role of galectin-3 in preclinical models of kidney injury. rIR leads to an increase of plasma and renal Gal-3 expression associated with acute tubular injury, promoting cytokine expression and immune cells recruitment. Toxic renal injury induces Gal-3 associated Th1 and Th17 recruitment for renal reparation and Gal-3 tissular associated PKcα cell apoptosis. Diabetic models induce an overexpression of Gal-3 via the Rpph1 pathway, promoting AGE downregulation via an upregulation of RAGE. In the cystic model, Gal-3 was expressed in cystic centrosome cilia and its inhibition aggravates the development of cyst. Gal-3 expression induces collagen formation via the endo180 receptor and the RAGE/ICAM pathway. Gal-3 is increased in plasma during type 3 cardiorenal syndrome leading to cytokine secretion, cardiac Gal-3-associated immune cells, and fibrosis induction.

**Figure 4 ijms-23-03124-f004:**
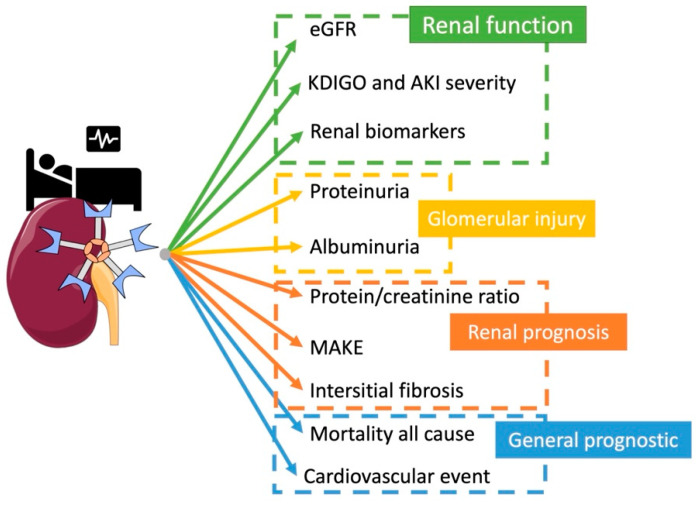
Kidney galectin-3 association with outcome in clinical studies.

**Table 1 ijms-23-03124-t001:** Inhibitors of Galectin-3.

	**Structure**	**Pharmacokinetic and Pharmacodynamics**	**Clinical Evidence**	**Administrative**	**References**
**CRD Based Multivalent**					
*Modified citrus pectin (MCP)*	Polypeptide formed with anhydro-galacturonic acid and galactose with shorter carbohydrate chains modified by pH and temperature	Gal-3 antagonist, soluble protein binding with Gal-3 CRD. Kd: 143 nM	Cancer: Phase II, single-center, open label, trial evaluating the safety and efficacy of MCP on PSA kinetics in prostate cancer (NCT01681823) Cardiac fibrosis: Phase III, randomized study, single-center trial evaluating the efficacy of MCP treatment to reduce cardiac fibrosis in patients with hypertension. (NCT01960946)	Approved by FDA, from Econugenics (Santa Rosa, CA, USA)	[45,46]
*GBC590/GCS100*	A combination of purified MCP (polymerized)	Gal-3 antagonist, soluble protein binding with CRD Unknown Kd.	Renal disease: -Phase I, open label study, evaluated the security of weekly doses of GCS-100 in patients with chronic kidney disease. (NCT01717248) -Phase IIa, placebo-controlled, randomized, single-blind study evaluated of weekly doses of GCS-100 in patients with chronic kidney disease and eGFR change. (NCT01843790) Cancer: -Phase II trials evaluated the reduction of metastasis and stabilized colorectal carcinomas during outcompeting Gal-3 in binding to its receptors. (NCT00110721)	Approver by FDA from La Jolla Pharmaceuticals (San Diego, CA, USA)	[47,48]
*RG 1–4*	Polypeptide formed with rhamnogalacturonan I (RG-I)-rich fragment	Link to Gal-3 CRD and limit Gal-3 oigomerization. Kd: 22 nM	No clinical data	No FDA approved, from Galectin Therapeutics (Norcross, GA, USA)	[46]
*Davanat and Belapectin*	A natural galactomannan polysaccharide	Multivalent binding with Gal-3 CRD Kd of Davanatt: 200–300 µM Kd of Belapectin: 2.9 µM	Liver fibrosis: Phases I, II, and III study in a multi-center, study, to evaluate the safety and pharmacokinetic of modified Davanat in subjects with non-alcoholic steatohepatitis (NASH) with advanced hepatic fibrosis to improve portal hypertension and oesophagial varice (NCT02462967, NCT04365868) Cancer: Phase Ib study of a galectin inhibitor (GR-MD-02) and ipilimumab in patients with metastatic melanoma (NCT02117362)	FDA approved, from Galectin Therapeutics (Norcross, GA, USA)	[44,49,50]
**Small-molecule carbohydrate-based inhibitors**					
*Lactose/LacNac or modified LacNac*	N-Aceetyl-D-lactosamine / or modified with a Arg144 guanidino group for modified LacNac	Natural ligant of Gal-3 with rich Galactomannan domain. Modified LacNac Kd: 320 nM	No clinical study and clinical implication as their affinity for Gal-3 are weak.	No FDA approval as a drug	[50,51]
*Small size synthetic monovalent inhibitor*	Thiodigalactoside scaffold (TD139/ GB0139 and GB1211)	Specific binding monovalent to subsite C and D of Gal-3 CRD GB0139 Kd: 2.3 nM GB1211 Kd: 30–55 nM	Pulmonary fibrosis: -Phase IIb, randomized, double-blind, multicenter, parallel, placebo-controlled study in subjects with idiopathic pulmonary fibrosis (IPF) investigating the efficacy and safety of GB0139. (NCT03832946) -Phase IIb, randomized controlled trial patient with HIV investigating the safety, tolerability and pharmacokinetic of TD139. (NCT02257177) -Phase IIb, single center, open label to assess the safety, tolerability and pharmacokinetics of GB1211 in participants with hepatic Impairment (Child Pugh B & C). (NCT05009680)	GB0139: FDA approved from Galecto inc (Ole Maaloes, Copenhagen, Denmark) GB1211: FDA approved from Galecto inc (Ole Maaloes, Copenhagen, Denmark)	[50,52,53]

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
