# Peer review of "Galectin-3 in Kidney Diseases: From an Old Protein to a New Therapeutic Target"

_ijms, 2022, doi:10.3390/ijms23063124_

Round 1

Reviewer 1 Report

In general, the review is well organized and covers all the aspects of the galectin-3 role in renal pathophysiology. However, there are a few concerns related to mistakes and misinformation found throughout the manuscript. 

1) Galectin-3 (Gal-3) is a 30KDa lectin protein.

Authors need to better define galectin-3, which is a lectin or a carbohydrate binding protein. A "lectin protein" does not sound correct. 

2) "carbohydrate recognizing domain (CRD)"

it should be replaced by "carbohydrate recognition domain (CRD)".

3) Galectins 4, 6, 8, 9 and 12 are composed of two united CRD (but each one is different).

By looking at the figure 1, one can assume that the two CRDs are identical.

4) Gal-3 is composed of a chimeric protein including one CRD domain

Galectin-3 IS a chimeric protein composed by one CRD and one regulatory N-terminal domain. Authors should better define galectin-3 structure.

5) Similar to lectins, Galectins  (??????)

Galectins are lectins. It is not known what the authors mean by saying that galectins are similar to lectins.

6) The CRD is a polypeptide fold binding domain with carbohydrates. Gal-3‘s CRD is able to bind multiple proteins, activating different signaling pathways [19]. 

Another sentence without proper meaning. Does the CRD contain carbohydrates? Is the CRD able to bind multiple proteins or carbohydrates?

7) Figure 2. Galectin-3 cellular role and localization within cells. 

The figure also shows galectin-3 on the cell surface.

8) collecting ducts ureteric metanephros bud branches

Another sentence without proper meaning.

9) Phase III, single-center, open label, trial evaluating the safety and efficacy of MCP on PSA kinetics in prostate cancer (NCT01681823).

After checking the clinical trials mentioned by the authors, I found several mistakes including that NCT01681823 is not phase III. In addition, some clinical trials began many years ago and either were not finished or the results were not reported. I puts some doubts on their importance and relevance, and the authors should present their critical opinion about these discrepancies. 

10)  modified citrulline pectin (MCP).

Authors should revise this name throughout the manuscript and correct to "modified CITRUS pectin".

11)  Grafted models are associated with immune cell-related graft dysfunction. Dang et 208 al. grafted BM13 kidney from WT mice in WT mice or in Gal-3 null mice. In WT mice, this 209 graft resulted in Gal-3 tissular and plasmatic upregulation. Gal-3 null mice had fewer tubular injuries, moderate fibrosis and less immune cells infiltration [74]. These two models confirmed the role of Gal-3 in the recruitment of immune cells during pathological context, and its inhibition improved renal outcomes. 

It is long known the role of galectin-3 in the recruitment of immune cells during inflammatory and immune responses. The authors should mention it.

12) Language mistakes should be revised.

Author Response

Comment of the reviewers (In bold)

We thank the reviewers for their careful reading and constructive comments that helped us to improve the quality of our study. We have addressed point by point all issues raised mainly by reviewer 1.

Reviewer 1

In general, the review is well organized and covers all the aspects of the galectin-3 role in renal pathophysiology. However, there are a few concerns related to mistakes and misinformation found throughout the manuscript. 

1) Galectin-3 (Gal-3) is a 30KDa lectin protein. Authors need to better define galectin-3, which is a lectin or a carbohydrate-binding protein. A "lectin protein" does not sound correct. 

To comply with the reviewer’s request, we replaced “lectin protein” by “lectin” (in the abstract or in page 1, lane 16) or “carbohydrate-binding protein” (page 1, lane 37)

2) "carbohydrate recognizing domain (CRD)" it should be replaced by "carbohydrate recognition domain (CRD)".

We apologize for this mistake that has been corrected in the revised version of our paper (page 1, lane 41).

3) Galectins 4, 6, 8, 9 and 12 are composed of two united CRD (but each one is different). By looking at the figure 1, one can assume that the two CRDs are identical.

We apologize for the confusion, we modified the figure and precise the type of CRD in the text (page 2, lanes 46-47).

4) Gal-3 is composed of a chimeric protein including one CRD domain. Galectin-3 is a chimeric protein composed by one CRD and one regulatory N-terminal domain. Authors should better define galectin-3 structure.

In the text, we revised the description of the Gal-3 structure (page 2, lanes 48-49).

5) Similar to lectins, Galectins  (??????) Galectins are lectins. It is not known what the authors mean by saying that galectins are similar to lectins.

We apologize for this mistake that has been revised (page 2, lane 51).

6) The CRD is a polypeptide fold binding domain with carbohydrates. Gal-3‘s CRD is able to bind multiple proteins, activating different signaling pathways [19].  Another sentence without proper meaning. Does the CRD contain carbohydrates? Is the CRD able to bind multiple proteins or carbohydrates?

We thank the reviewer for highlighting this point. We revised our text to make it more clear (page 3, lane 65-68).

7) Figure 2. Galectin-3 cellular role and localization within cells. The figure also shows galectin-3 on the cell surface.

We fully agree with the reviewer’s comment. The title of figure 3 was modified (page 4, lane 102).

8) Collecting ducts ureteric metanephros bud branches…..another sentence without proper meaning.

We apologize for the mistake. The sentence has been modified (page 4, lane 118).

9) Phase III, single-center, open label, trial evaluating the safety and efficacy of MCP on PSA kinetics in prostate cancer (NCT01681823). After checking the clinical trials mentioned by the authors, I found several mistakes including that NCT01681823 is not phase III. In addition, some clinical trials began many years ago and either were not finished, or the results were not reported. I put some doubts on their importance and relevance, and the authors should present their critical opinion about these discrepancies. 

We thank the reviewer for this comment. The clinical trial NCT01681823 is in fact at phase II but written as phase III in ClinicalTrial.gov. We modified this point and discussed the importance and relevance of these studies (page 4, lanes 109 -113).

10)  Modified citrulline pectin (MCP). Authors should revise this name throughout the manuscript and correct to "modified CITRUS pectin".

This point has been corrected in our revised version (page 7, lane 224).

11)  Grafted models are associated with immune cell-related graft dysfunction. Dang et 208 al. grafted BM13 kidney from WT mice in WT mice or in Gal-3 null mice. In WT mice, this 209 graft resulted in Gal-3 tissular and plasmatic upregulation. Gal-3 null mice had fewer tubular injuries, moderate fibrosis and less immune cells infiltration [74]. These two models confirmed the role of Gal-3 in the recruitment of immune cells during pathological context, and its inhibition improved renal outcomes. It is long known the role of galectin-3 in the recruitment of immune cells during inflammatory and immune responses. The authors should mention it.

As suggested by the reviewer the above-mentioned point has been added in the revised version of the manuscript (page 7, lanes 228 -238).

12) Language mistakes should be revised.

We apologize for these mistakes that have been corrected in the revised version of our paper.

Reviewer 2 Report

Well written and organized comprehensive review describing the role of Galectin in pathways involving renal damage and fibrosis and its role as a biomarker. The review is well organized with just the right amount of discussion about Galectin and the various pathophysiological pathways it is involved in related to renal function. 

Author Response

Reviewer 2

Well written and organized comprehensive review describing the role of Galectin in pathways involving renal damage and fibrosis and its role as a biomarker. The review is well organized with just the right amount of discussion about Galectin and the various pathophysiological pathways it is involved in related to renal function. 

We thank the reviewer for his/her enthusiasm and positive comments.

Round 2

Reviewer 1 Report

Line 71 and 72: The authors claim that the N-terminal "domain regulates Gal-3 oligomerization or Gal-3 monovalent secretion". Regarding gal-3 secretion, so far, it is not known by which mechanism gal-3 is secreted to the extracellular milieu. Please remove this affirmation.    Line 85: The authors state that "Gal-3 has a membranous form to modulate the interaction between epithelial cells and extracellular matrix or with other cells". Please revise since "membranous form" is not a "state" of galectin-3. Gal-3 is found in the plasma membrane of cells where it interacts with glycosylated ligands (proteins, lipids, etc)... 

Author Response

We thank the reviewer once more for the constructive comments. 

We have addressed both issues (in red) in the new revised version of our manuscript.

We hope that our review will now be acceptable for publication.
